# Optimizations for Passive Electric Field Sensing

**DOI:** 10.3390/s22166228

**Published:** 2022-08-19

**Authors:** Julian von Wilmsdorff, Arjan Kuijper

**Affiliations:** 1Fraunhofer IGD Darmstadt, Fraunhoferstraße 5, 64283 Darmstadt, Germany; 2Department of Computer Science, Technical University of Darmstadt, 64289 Darmstadt, Germany

**Keywords:** capacitive sensing, passive electric field sensing, sensors, signal processing

## Abstract

Passive electric field sensing can be utilized in a wide variety of application areas, although it has certain limitations. In order to better understand what these limitations are and how countervailing measures to these limitations could be implemented, this paper contributes an in-depth discussion of problems with passive electric field sensing and how to bypass or solve them. The focus lies on the explanation of how commonly known signal processing techniques and hardware build-up schemes can be used to improve passive electric field sensors and the corresponding data processing.

## 1. Introduction

Passive electric field sensing is a sensing technique related to classical electric field sensing technologies such as loading mode or transmit mode sensors. As its name already suggests, passive electric field sensing does not, in contrast to classical capacitive sensors, emit an electric field on its own, but rather uses naturally occurring fields to gain knowledge about the environment. J. R. Smith provides a detailed explanation of different (active) electric field sensing methods [1], while a good mathematical model for passive electric field sensing can be found in [2]. A setup for passive electric field sensors was, among others, published by Prance et al. [3].

The basic principle of this kind of sensors is explained in Section 3. Many objects in our environment are able to carry an electric charge on their surface. This holds true for humans as well, a fact that we are painfully reminded of every time we experience a shock while touching the door of a car or another conductive surface. These charges on our skin or on the surface of the object that we want to detect attract the opposite charge in objects placed in our surroundings. This means that when moving a hand over a piece of wire, a current is induced in it. It is important to note that a charged and stationary object near this peace of wire do not induce any kind of current. Because the induced currents when moving one’s hand near a wire are very small, an impedance converter, such as discussed later on in Section 3, is needed. This current that a user generates in the electrode can be measured with respect to the input impedance of the operational amplifier as a voltage.

There are different approaches to optimizing passive electric field measurements. Before discussing possible optimizations, however, it should be clarified that optimization in this context can imply several things. The most important aspect is to enhance the signal to noise ratio of a sensor, which can increase the measurement range of a sensor as well as its sensitivity. Other aspects that can be improved are the reliability and validity of a sensor, as these do not increase when enhancing the signal to noise ratio.

All considerations concerning hardware optimizations in this paper are carried out with the Linoc toolkit [4]. All of these considerations are transferable to other passive electric field sensors, because these considerations represent the implementation of common techniques for passive electric field sensors and these common techniques are already used in similar forms today in different fields of applications. For this reason, basing the following optimizations on a specific sensor toolkit does not impact the validity of the presented propositions.

## 2. Signal Processing

As described by vone Wilmsdorff et al. in [5], the detection of local extrema is an important method of gaining information from electric field data, baselined by using hardware components such as high valued resistors to slowly pre-charging the electrodes. Peak detection can be much more challenging for systems which lack this kind of baseline generation, because these sensors reach their saturation limits before an action is completely recorded. Even systems with integrated hardware baseline compensation can run into this kind of problem, as shown for example in the use of passive electric field sensors for traffic observation [6], when events involving fast-moving objects that can carry a lot of charge must be recorded.

Even if an event does not saturate the sensor, and even if the electronically generated baseline for this signal would be a precise predefined value, the absolute level of the recorded sensor signal most likely does not transport any information, as the amount of charge on an object changes over time. For example, if two simple step patterns of a person that walks by a passive electric field sensor are recorded, the absolute level of the sensor amplitude can differ beyond the sensor’s noise. The more time lies between these two recordings, the higher the potential amplitude difference.

As a matter of fact, not only can the amount of charge on an object change, the algebraic sign of the amount of charge can change as well, seemingly flipping the recorded sensor signal around the time axis.

This leaves the question of whether the amplitude of an electric field sensor is even able to contribute information to any use-case. In many use-cases, a circumstance that can be exploited to gain information is that the amount of charge changes only slowly over time. Note that in this general discussion it is not possible to clearly quantify the term “slowly” more exactly, as it is dependent on the use case of the technology. In a use case scenario in a bathroom, for example, by touching grounded faucets or valves the user nearly instantly discharges. A real-life recording of this kind of event can be seen in Figure 1.

The picture illustrates a discharge event of a person touching an electrical ground while being near a sensor, recorded with a passive electric field sensor configured with a 50 Hz sampling rate. It can be observed that this discharge event is shorter than the 20 ms resolution of the used sensor and that the sensor needs approximately 150 ms to re-calibrate itself to reach its normal baseline again.

The described scenario and the recorded data show that comparing absolute voltage levels is not applicable if the use case is prone to rapid charge and discharge events. However, in scenarios where this is not the case, a comparison between sensor amplitudes can make sense. An example use case is provided in [5]. Here, amplitudes are compared in the event of a person entering or exiting through a door equipped with a passive electric field sensor. While entering or exiting, there are no opportunities for a discharge event to occur. Even if the person in question wore no shoes, or shoes that grant an ohmic connection to ground, this statement nonetheless holds true, as the discharge would take place before reaching the door. Furthermore, the time span of traversing a door frame is short, and thus that the charge gained via the triboelectric effect during this event concerning the shoes of a person is negligible.

In summary, to gain the most information out of the absolute level of charge, the following actions are recommended:The use case in question should provide no opportunities for the measurement target to gain or lose large amounts of charge.While the algebraic sign of the charge has to be ignored, it could be of use when comparing short periods of the same event.When comparing amplitudes of different sensors, an initial normalization and baseline calculation are required due to variations in electrical components.

Regarding the creation of a baseline, several algorithms are explored within the scope of this work. Before analysing different algorithms, however, we discuss the properties of a signal measured by a passive electric field sensor.

Nearly all recordings shown in this work are sampled at a sampling rate of 50 Hz. This is because in Europe the operating frequency for power grids indoors is 50 Hz. When sampling a 50 Hz sine wave with exactly 50 Hz, the output function is constant. This holds true even when the sine wave is sampled with an initial delay, that is, with a phase shift greater than zero. Figure 2 shows such a 50 Hz sine wave that was sampled with an exact 50 Hz sampling frequency.

The advantage of this technique is that very slow ADC sampling rates can be used. As modern ADCs support sampling rates up to several gigasamples per second, as shown by Greshishchev et al. [7], it is possible to acquire a large number of samples in this period of time. This enables the use of low pass filters to improve signal quality. A simple implementation of a low pass filter with multiple ADC samples is to average all collected additional samples.

Another advantage of the undersampling approach is that it is very easy to carry out signal processing after the measurement. In applications where an absolute value threshold for the detection of events is used, signal processing can even be skipped completely without the need to filter distortions from 50 Hz power lines. For many other applications, it is then sufficient to remove the steady component of the signal, for example, by computing the derivative of the original curve. The derivative of a discrete signal is a very inexpensive preprocessing technique in computational terms. This makes it ideal for use with passive electric field data. In comparison to complex filters that remove the steady component, the derivative can be computed live for every new measured point of data, which is not possible if data first have to be buffered. The derivative for the nth value is computed by
(1)$$\Delta f_{n}=f_{n}-f_{n-1}.$$
This involves a single operation per value (function complexity O(n)) and a single integer to be stored (the value $f_{n-1}$). These low requirements, combined with the fact that many events recorded with a passive electric field sensor are short peak events rather than smooth curves (see Figure 1), make the derivative an ideal pre-processing technique.

Problems with the undersampling technique arise if the frequencies of the analog-to-digital converter and that of the interfering field are different. Even a slight difference in one of these frequencies results in an aliasing effect, as depicted in Figure 3.

The 50 Hz sine wave was sampled with a slightly lower frequency than 50 Hz, resulting in a low-frequency sine wave as the sampled result. Because no clocks are absolute, no ADC can achieve a perfect result. Furthermore, the frequency of the electric power grid, the main source of disturbances when sampling passive electric field data indoors, is not constant and ranges between 49 Hz and 51 Hz. This is the reason for the aliasing effects shown in Figure 3 when using undersampling with no further filtering, and is inevitable.

The formerly discussed preprocessing technique for removing the steady component of a passive electric field signal by calculating the first derivative can be used for measurements containing aliasing effects as well. This places applications that involve the analysis of zero crossings of the measured signal by these kinds of disturbances at a disadvantage. As shown in [5], the detection of zero crossings for passive electric field data can play a crucial role in improving the detection of events. Additional preprocessing steps have to be implemented in order to compensate for the falsely generated zero crossing of aliasing effects, for example, making sure that the signal first reaches a certain threshold before a zero crossing is marked as valid.

A different approach is to use more suitable baseline algorithms. Fast Fourier Transformation, first proposed in the 1965 [8] and one of the oldest signal processing algorithms still used today, is a fitting method that can process passive electric field data. Advantages of this algorithm include the low computing power ($O(n\log_{}\left(n\right))$, compared to the $O(n^{2})$ needed by the original Fourier Transformation) required to perform the calculation and the fact that several specific unwanted frequencies can be filtered at once. For passive electric field data recorded by the Linoc toolkit, these frequencies are the steady component and very low frequencies that can occur because of aliasing effects (0 Hz to 1 Hz), as well as frequencies over approximately 15 Hz, since human movement is located in this very low frequency domain. A disadvantage of Fast Fourier Transformation is the fact that it requires a buffer of measurements. This means that the algorithm has a fixed delay for incoming data. While this does not harm its ability to produce real-time results, it means that an insensitive time exists whenever Fast Fourier Transformation is used. This complicates the “online usage” of this algorithm (e.g., showing a filtered live plot of a sensor to be able to immediately observe the data). Another disadvantage is the buffer itself, which can require a large amount of RAM that might not be available on embedded systems with micro-controllers lacking the proper amount of system memory.

This leads us to two simpler approaches to implementing baselines for passive electric field data that require very little RAM and have a runtime behaviour of O(n), which is even smaller than that of Fast Fourier Transformation. These baselining approaches are therefore perfectly suited to implementation on embedded systems.

The first approach for creating a baseline with a passive electric field sensor as an input modality is to use paired slow-following and a fast-following functions in the form of exponentially moving averages. The slow follower follows the sensor inputs more passively than the fast follower. Both functions can be derived as follows:(2)y(n)=x(n)·s+y(n−1)·(1−s)
where x(n) is the *n*-th sensor value, y(n) is the corresponding output of the follower function, and s∈[0,1] is the speed or smoothness factor of the follower function. Higher values for *s* mean a stronger tendency to follow the original sensor output, making it more prone to high frequency distortions. In the case where *s* equals 1, the follower function matches the original input exactly, whereas if *s* equals 0, the follower function remains a constant number. An initial value for y(0) has to be provided, and should be chosen in the region of the presumed baseline for faster convergence, although y(0) can be chosen arbitrarily.

Figure 4 depicts how a noisy signal can be smoothed and baselined using two different exponential moving averages. The original signal shown in Figure 4a exhibits a function with noise and a slowly rising baseline drift. Ideally, the calculated baseline for this signal would be a straight line with the same slope as the occurring baseline drift. Figure 4b,c are both exponential moving averages, but with different smoothing factors to generate a function that follows the original signal slower and faster respectively. The result, as presented in Figure 4d, is the subtraction of the slow following function from the fast following function. It can be noted that the resulting function is less noisy than the original signal. In addition, the original upward trend is negated, leading to a function with a baseline centered around zero.

In the example shown in Figure 4, the initial values of both the fast and slow following functions are set to zero in order to demonstrate the convergence behaviour of the exponential moving average functions. Because of the time needed for both functions to converge sufficiently against the original signal, the result in Figure 4d includes a distortion at the beginning. This effect can be neglected, as a simple solution to this problem is to choose the first measured value of the original signal as the initial value for both exponential moving averages, eliminating the extra time to convergence.

Because of its exponential nature, this kind of baseline approach follows deflections of a curve that are not part of a baseline drift, and are instead part of the useful signal to be extracted. This unwanted behaviour can be mitigated by replacing the first summand of Equation (Equation 2) with a constant. As this means that the baseline generated by this new equation can only grow in one direction, a few more modifications have to be implemented.

Figure 5 shows the complete approach to limiting the growth of the baseline. The generated baseline grows with a maximum of a constant *c* towards the original signal per measurement taken. This positive non-zero constant can be chosen according to the needs of the application. Hence, a larger value of *c* should be chosen in a noisy environment in order to compensate for the larger baseline deviations. The consequence of using this algorithm compared to an exponential moving average is that the baseline filters out part of the wanted signal, however, instead of removing a percentage value of the wanted signal, the loss is now limited to an absolute value per measurement.

In contrast to Figure 4, in Figure 5 the slow-following baseline is replaced by a baseline generated by the algorithm. Again, for the sake of demonstrating the convergence behaviour the initial value of the first baseline was set to zero instead of to first value of the original signal (as demanded by the algorithm shown in the first step of Figure 5).

As can be observed by comparing Figure 6b to Figure 4b, the generated slow-following baseline converges less in the middle of the signal, where a larger amplitude is visible. On the other hand, it converges more strongly in parts of the signal with smaller amplitudes. This indicates that this approach for calculating a baseline for a passive electric field sensor is more suitable in situations with strong oscillating signals and less suitable for signals of a more delicate nature.

Another disadvantage of generating a baseline with a limited slope is the existence of scenarios in which the difference between the calculated baseline and the signal does not decrease, or even increases (divergence). While this is the normal and desired behaviour of a baseline for sections of the signal with high deflections, it is a problem when it occurs over longer periods of time, as this indicates that the baseline slope is smaller than any potential baseline drift.

After separating the useful signal from the steady component with the discussed techniques, the absolute values of the amplitudes can be compared. As already described in the preceding sections, an immediate comparison of amplitudes can make sense in scenarios where discharge of the object of interest is unlikely.

## 3. Physical Optimization

There are several considerations about the physical design of a sensor that have to be taken into account when using passive electric field sensing for different use cases. As shown in Figure 7, a resistor is used in the structure of the passive electric field sensor presented by [9,10]. As already mentioned, these resistors are responsible for stirring the baseline of the sensor towards a desired predefined value which is defined by the voltage in front of these resistors. $R_{bias}$ behaves like a very slow pull-up resistor because, as discussed in [10], it was set in the region of gigaohms.

Because both values of $R_{bias}$ are smaller than the input impedance of the operational amplifier, these resistors are the dominant factor of the circuit shown here when it comes to measurement range. This is because the small current that is induced in the electrode of the sensor when an object is passing by cannot be picked up by the amplifier if it is overwritten beforehand by the baseline current of $R_{bias}$. However, increasing the resistance of $R_{bias}$ or even removing it, that is, $R_{bias}\to \infty$, can render the sensor useless because its tendency to run into a saturation scenario increases with the decrease of the baseline current.

This means that in an optimal laboratory environment where the induced current on the electrode is known, the selected value of $R_{bias}$ should be as large as possible (in order to increase the measurement range), yet small enough that the sensor never reaches full saturation (to ensure maximum information gain). This idea can be a step further and the fixed value of $R_{bias}$ replace with an electronically controlled potentiometer. The biggest problem with this design is the unavailability of potentiometers within the needed ohmic range, as potentiometers or adjustable resistors with values over 10 MΩ are highly unusual. As shown in [10], setting up a passive electric field sensor with a measurement range of approximately two meters requires resistor values in the gigaohm range.

It should be taken in contemplation that lowering the value of these resistors further can be useful to save steps in the signal processing afterwards even if the sensor is not prone to running into saturation. Even if there are no saturation issues in the currently relevant use case, increasing the baseline current can dampen aliasing effects such as those shown in Figure 3, as well as other external distortions of the signal. It should be borne in mind that this technique lowers the amplitude of the detected signal. Hence, suppressing distortions with an increased baseline current is only applicable if the use case generates a signal that is strong enough to compensate for this artificially-created loss in signal strength.

Instead of enhancing the measurement range by increasing the resistors with respect to the baseline current, another optimization technique that can be used is to simply enlarge the electrodes of the passive electric field sensor. Increasing the area of the electrodes increases the capacitive coupling between the electrode and the object of interest, and hence increases the displacement current generated by moving objects, leading to an improvement in the signal to noise ratio of the sensor. An enlargement of the electrodes is of course only feasible in applications where the spatial dimensions of the measurement apparatus are not a concern, which eliminates mobile and embedded applications.

Other disadvantages of an enlarged electrode are similar to the disadvantages of setting the value of the baseline resistor too high. As previously discussed, the baseline current is relevant because it prevents saturation of the passive electric field sensor. An electrode with its size set too high means that the baseline current is not able to compensate for the strong currents generated by it, which again leads to the saturation of the sensor and ultimately to the loss of information.

A more obvious method for passive electric field sensors with improved signal to noise ratio is to reduce the distance to the measured object. The two previously discussed measures (enlargement of the electrode and adjustment of the pull-up resistors for the baseline) both have an effect on the measured displacement current. This holds true when reducing the distance to the measured object as well because the capacity of the formed capacitor is antiproportional to its distance of the electrodes, as depicted by the equation for the capacitance of a capacitor,
(3)$$C=\epsilon_{0}\epsilon_{r}\frac{A}{d},$$
where *C* is the electrical capacity, $\epsilon_{0}\epsilon_{r}$ is the permittivity of the material in between the capacitor plates, *A* is the area of the capacitor plates, and *d* is their distance. Hence, the displacement current can be increased by lowering the distance from the sensor to the object in question. Again, as with enlarging the electrodes, this method of increasing the signal to noise ratio is highly dependent on use case and might not be applicable in scenarios where the objects of interest are strongly varied in their distance to the sensor.

Another consideration when setting up a passive electric field sensor is the shaping of the electrodes and their surroundings. Prance et al. showed that leakage currents can be reduced to femto amperes [3] via efficient shielding. Shielding does not reduce the noise of the sensor itself, only the noise picked up from the environment. This means that the region of interest can be selected more precisely by narrowing the “field of view” of the sensor, as demonstrated in [5].

Because electrical field lines always enter and exit the surface of an electrode perpendicularly, concentrating them is possible. In addition to the efficient use of shields, Prance et al. showed an example of how to focus the electrode of a passive electric field sensor using a needle with a 50 µm tip [11]. A different approach is to couple the electrode to a different object, which henceforth acts as the electrode, instead of forming the electrode on the sensor itself. Of course, the object that the sensor is attached to must have conductive properties in order to properly act as an electrode. This has been demonstrated in [6], for example, where the sensor was installed on a whiteboard and then used to detect a person touching the whiteboard.

## 4. Summary

Several techniques to optimize the use of passive electric field sensing have been discussed in this paper, including signal processing and hardware considerations.

Regarding the different signal processing techniques, we have discussed which parts of a passive electric field signal transport useful information and which parts do not and adapted known signal processing techniques for utilization in the context of passive electric field sensing. This comprises baseline calculation, signal smoothing, and considering the complexity of the used algorithms.

In addition to the algorithmic part of the signal processing, a description of which parts of a passive electric field signal can be used under which circumstances has been provided. With respect to hardware optimization methods, we have discussed which sensor assembly groups can potentially increase the signal to noise ratio of the measured signal, including baseline resistors, electrode size and distance, and shielding of a passive electric field sensor.

## 5. Conclusions and Future Work

In summary, many optimization methods for passive electric field sensors are highly dependent on the application for which they are used. We have discussed cases in which increasing the sensitivity of a sensor is not necessarily beneficial to the quality of measurement results due to resulting saturation problems in the hardware.

The same is true for software optimizations. While the complexity of filter algorithms implemented directly on embedded devices, including sensors themselves, is increasing nowadays thanks to higher overall micro-controller performance, the choice of filter remains very much dependent on the use case. As discussed, this is a consequence of noise in the environment in which the sensor is to be used.

These considerations are currently leading to new areas of research, such as filters implemented directly in hardware for passive sensing of electric fields. While hardware signal filters are a topic that has been studied for decades and in various aspects, implementation for passive sensing of electric fields poses a new challenge. As shown in Figure 7, such filters need to be implemented without active components in order to avoid the effects of nonlinearity, and must avoid affecting the high input impedance of the sensor.

## Figures and Tables

**Figure 1 sensors-22-06228-f001:**
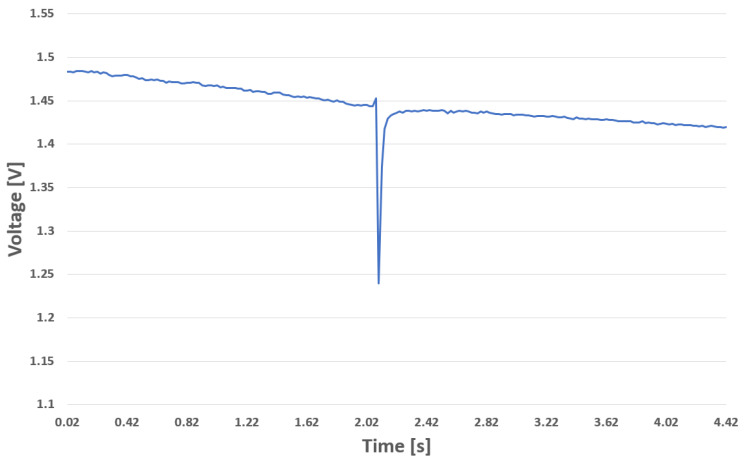
Discharge event.

**Figure 2 sensors-22-06228-f002:**
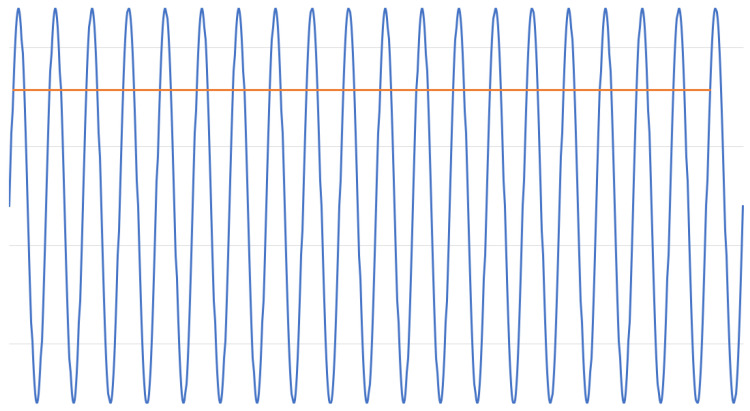
A 50 Hz sine wave sampled with 50 Hz and the resulting function.

**Figure 3 sensors-22-06228-f003:**
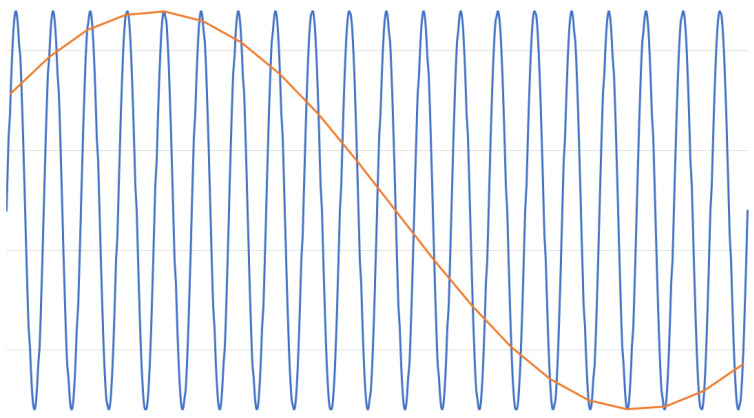
Aliasing effect shown on a 50 Hz sine wave.

**Figure 4 sensors-22-06228-f004:**
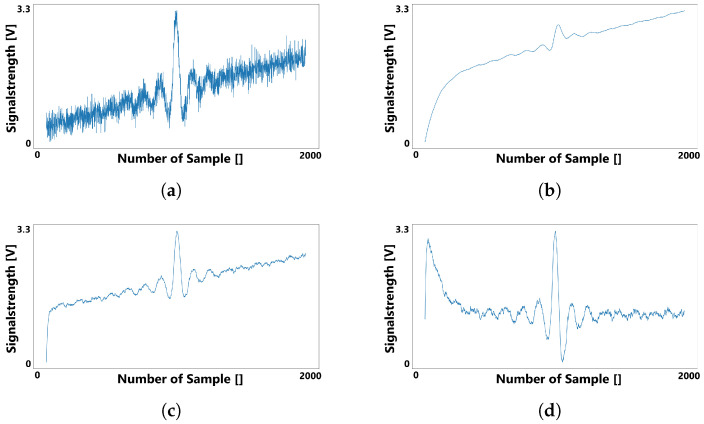
Smoothing a signal with two exponential moving averages: (**a**) depicts the original signal, while (**b**) (follow-factor s=0.99) and (**c**) (s=0.9) present two moving averages with different following factors applied to the original signal. Subtracting these results in the preprocessed signal shown in (**d**).

**Figure 5 sensors-22-06228-f005:**
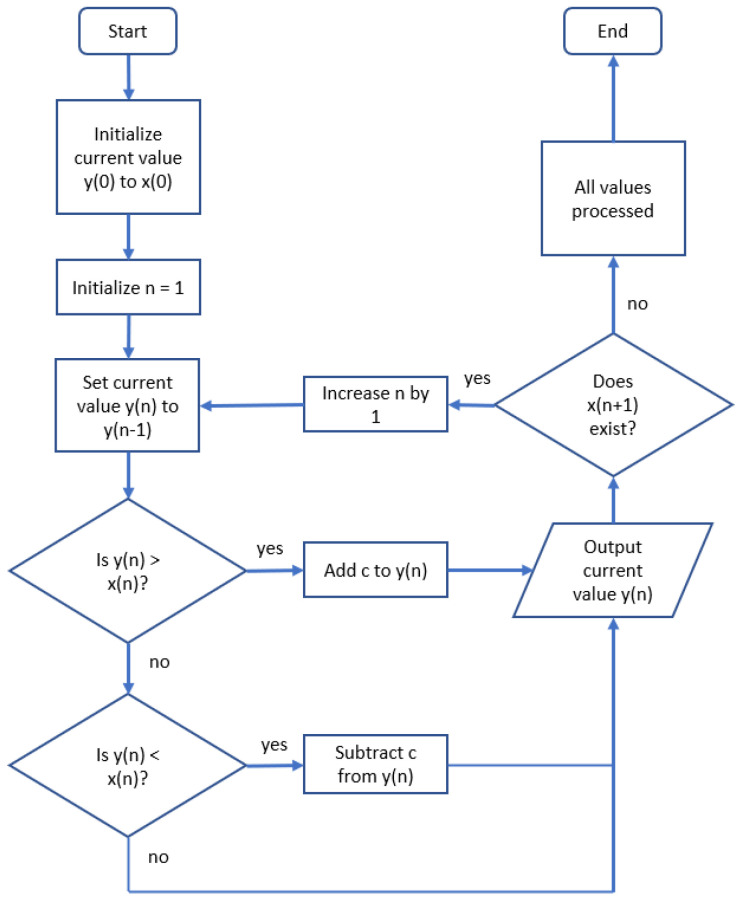
Linear baseline algorithm.

**Figure 6 sensors-22-06228-f006:**
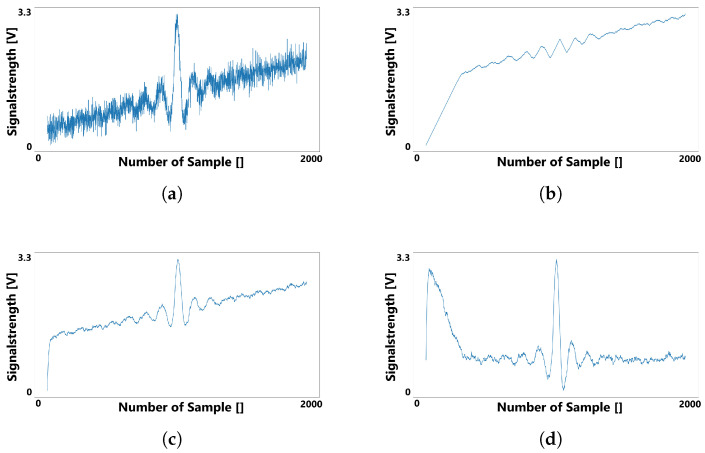
The noisy signal (**a**) smoothed using an exponential moving average (**c**) (with s=0.9) and a linear baseline (**b**), resulting in the signal depicted in (**d**).

**Figure 7 sensors-22-06228-f007:**
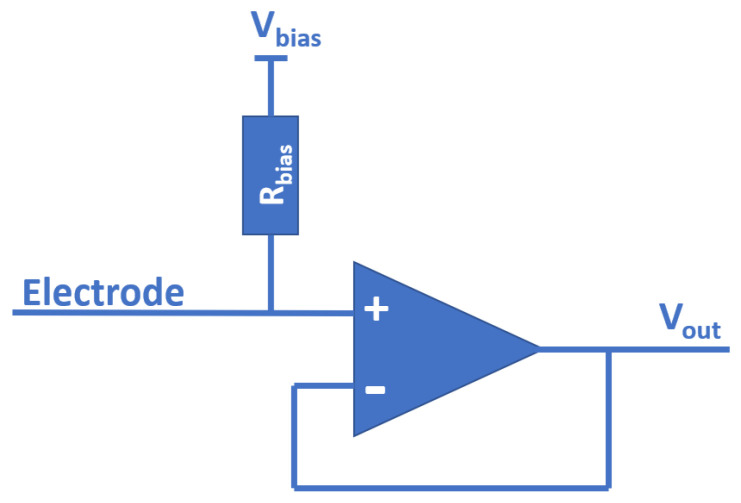
Simplified passive electric field sensor.

## Data Availability

Not applicable.

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
