# Peer review of "Optimizations for Passive Electric Field Sensing"

_sensors, 2022, doi:10.3390/s22166228_

Round 1
Reviewer 1 Report
The authors present a study that explores an in-depth discussion of problems with passive electric field sensing and how to bypass or solve those. The author's focus lies on an explanation of how commonly known signal processing techniques and hardware build-up schemes can be used to improve passive electric field sensors and the corresponding data processing.
The paper is well written, but in my opinion, it lacks a better description of the state of the art and a better explanation of the novelty of this work. Especially, the authors should compare their linearity, error, performance, and resolution results with the literature in order to demonstrate their contribution.
Please improve all the figures, in their present form, they are of very poor quality.
The captions of Figures 4 and 6 should include all descriptions of items a), b), c) ....
Please define "SNR"
This is my major concern regarding this work. At this step, authors should compare their work with other ones in other to demonstrate the novelty of their work.
Please add a conclusion and future work.
What benefits does your hardware have in terms of energy consumption?
Author Response
Thank you for your feedback. Unfortunately, it is very difficult to compare things like linearity, error, performance, and resolution of sensors that are known from other literature in a general discussion paper, especially when most of the optimizations discussed are very use-case dependent. I tried to give a better explanation in the introduction of the discussed technology in an improved version of this paper so that the reader gets a better idea of how certain enhancements will affect the technology.
You also have been pointing out clear weaknesses of this work which will be addressed. The term "SNR" stands for signal to noise ratio. In the next version of this paper, i have replaced this abbreviation to get rid of the issue.
You were also absolutely right about the quality of the images. I have revised ALL the pictures to make them more readable, not just figures 4 and 6.
I also added a short conclusion and future work section at the end as you suggested, in which i also tried to point out the contribution a little bit more.
I am not sure what to respond to your question concerning the energy consumption of the used hardware, since energy consumption was not a focus of this work.
Thanks for sharing your thoughts on this work in such a constructive way!
Reviewer 2 Report
11. The author needs to double-check the format of references, spelling, and capitalization mistakes in the manuscript.
22. The captions of figures 4, and 6 need to place below the figures.
33. The figure label is too small and can’t see anything. The author needs to format them.
44. In figure 5, what is the constant “c”? (value, range of the constant “c”?)
55. What is the practical application of the optimization process?
Author Response
Thank you for your feedback. I will revise the text for spelling mistakes for the next version by asking a native speaker to help me out.
You were absolutely right about the quality of the pictures. I reworked all of the pictures in the paper for better readability.
Regarding the practical application of the discussed optimizations: These are very use-case depended and discussing all practival applications is worth a paper on its own. This is why i described in the introduction that the term optimization can refer to several things, including better signal to noise ratio of a sensor or better reliability and validity.
I also improved the description of the constant c of the presented algorithm as you suggested.
Thank you for giving me such a constructive feedback!
Reviewer 3 Report
Dear Authors, even though the topic is really not in my field of expertise the abstract sounded interesting so I was looking forward to read the paper. Unfortunately, in my opinion, the abstract does not really reflect what is in the paper (or the paper did not deliver what I expected..), or at least it is a bit confusing.
In the first part of the paper there is a passage about sampling (dealing with sampling, undersampling and oversampling) and in my opinion it is really a big mess. The statement "Figure 2 shows such a 50Hz signal that was perfectly sampled with a 50Hz sampling frequency" is really very wrong. And not only this particular sentence. I am missing a concept that would provide clear path to the reader from “standard” sampling (fulfilling the Nyquist theorem) to possible undersampling for a band-pass limited signal (which I think is not this case).
There are discussed some digital signal processing techniques but without saying first what is the overall goal.. (some extraction of features with specific frequency band given by the events to be detected)
The part concerning the Rbias (Figure7) is also unfortunate - the paper does not mention parameters of the op-amp (or type of an op-amp / bipolar-cmos-jfet) that influences the Rbias value (input currents ). There is an “instrumentation amplifier” mentioned although there is an operational amplifier used - big difference between those two..
It is almost impossible to read values in graphs on Fig.4, but also the others should made bigger..
Author Response
Thank you for your reply. The first part of the paper that includes the mentioned section of sampling and under-sampling is meant to point out the advantages and disadvantages of using under-sampling in regard to a technology that is prone to "noise" from 50Hz powerlines. Discussing the technique of sampling itself is not meant to be a focus of this work. Yet you are absolutely right that the statement you pointed out is wrong. I will correct this issue.
Same goes for the use of the word "instrumentation amplifier". There are versions of this sensor using an instrumentation amplifier, but as you already pointed out, the presented simplified version is using an operational amplifier. Thank you for pointing out this issue!
I also reworked all the figures to improve their readability, not only figure 4.
Thank you for giving me feedback in such a constructive way!
Round 2
Reviewer 1 Report
I appreciate the authors' answers to my comments and suggestions. However, I still have some concerns about the previous review graphics style.
In figures 4 and 6, in the caption of the figure, I suggest that you carry the details of each of the sub-figures (while in the sub-figure you should only keep only the letter). For example in figure caption 4, "Figure 4 Smoothing a signal with two exponential moving averages. (a) Noisy signal. (b) Slow following exponential moving average, s = 0.99. (c) Fast moving average, s = 0.9 and (d) subtracted slow & fast follower."
Author Response
I fixed the graphics style according to your suggestions.
Best regards!